# `syftr`: Pareto-Optimal Generative AI

**Alexander Conway**[1,*]  **Debadeepta Dey**[1,*]  **Stefan Hackmann**[1,*]  **Matthew Hausknecht**[1,*]
**Michael Schmidt**[1,*]  **Mark Steadman**[1,*]  **Nick Volynets**[1,*]

[*]Names in alphabetical order by last name.
[1]DataRobot

**Abstract**  Retrieval-Augmented Generation (RAG) pipelines are central to applying large language models (LLMs) to proprietary or dynamic data. However, building effective RAG flows is complex, requiring careful selection among vector databases, embedding models, text splitters, retrievers, and synthesizing LLMs. The challenge deepens with the rise of agentic paradigms. Modules like verifiers, rewriters, and rerankers—each with intricate hyperparameter dependencies have to be carefully tuned. Balancing tradeoffs between latency, accuracy, and cost becomes increasingly difficult in performance-sensitive applications.

We introduce `syftr`, a framework that performs efficient multi-objective search over a broad space of agentic and non-agentic RAG configurations. Using Bayesian Optimization, `syftr` discovers Pareto-optimal flows that jointly optimize task accuracy and cost. A novel early-stopping mechanism further improves efficiency by pruning clearly suboptimal candidates. Across multiple RAG benchmarks, `syftr` finds flows which are on average $\approx 9\times$ cheaper while preserving most of the accuracy of the most accurate flows on the Pareto-frontier. Furthermore, `syftr`'s ability to design and optimize also allows integrating new modules, making it even easier and faster to realize high-performing generative AI pipelines. Code

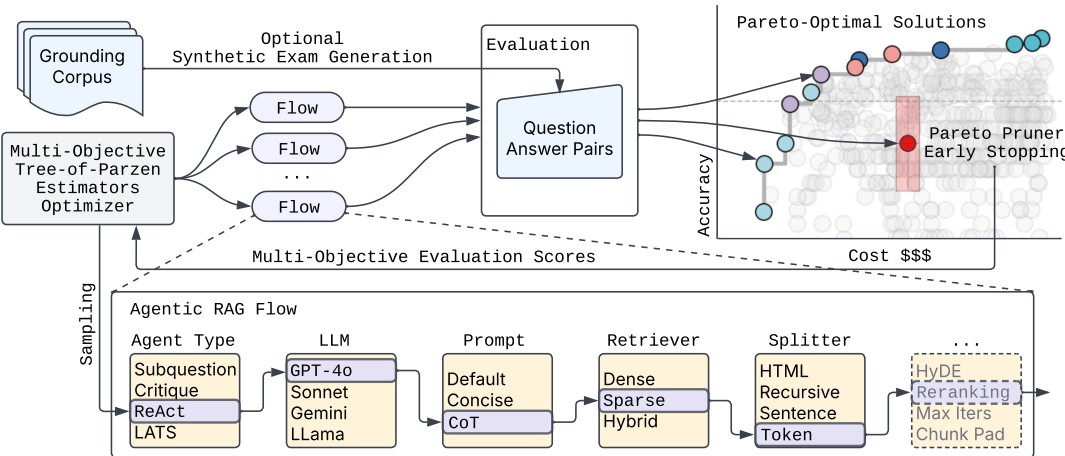

Figure 1: Given a grounding corpus, `syftr` searches over more than $10^{23}$ unique RAG flows to find a Pareto-frontier (optimal tradeoff curve) between task accuracy and cost.

## 1  Introduction

Recent advances in large language models (LLMs) have significantly expanded their capabilities across a range of linguistic tasks. However, they still suffer from limitations such as hallucinations,

outdated knowledge, and poor grounding in factual or domain-specific information [1, 2]. Retrieval-Augmented Generation (RAG) addresses these challenges by dynamically integrating external knowledge into model outputs, improving accuracy and reliability by grounding responses in verifiable sources [3, 4].

To operationalize RAG, generative AI flows (or pipelines) orchestrate how LLMs interact with external or evolving data—either through vector database retrieval or by directly injecting content into large context LLMs' prompts. These flows can range from static, imperative designs to dynamic, agentic systems that adapt at runtime. A growing ecosystem of frameworks—such as LangChain [5], AutoGen [6], Haystack [7], CrewAI [8], and LlamaIndex [9]—has emerged to support rapid prototyping and deployment of such applications. Yet, with this abundance of tools and design choices, developers face increasing complexity in selecting optimal configurations.

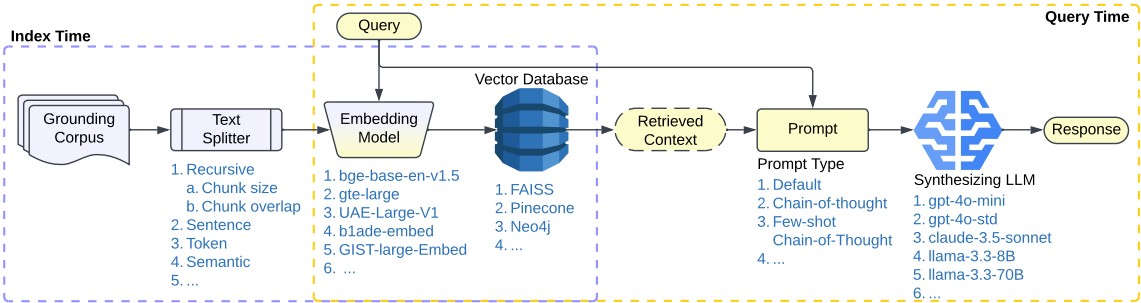

Figure 2: In this simplified view of the canonical RAG flow we term as "vanilla RAG", the developer has many choices for text splitter, embedding model, vector database, prompt type, and synthesizing LLM. We show a few choices to elucidate the point but there are far more choices available for each of these modules leading to hundreds of unique vanilla RAG flows each with different latency, accuracy, and cost tradeoffs.

To illustrate this challenge, we begin with the simplest RAG flow, illustrated in Fig. 2, which we refer to as the "Vanilla RAG" flow [1]. A grounding dataset is split into chunks using a text splitter, embedded into vectors via an embedding model, and stored in a vector database (VDB). At query time, the user query is embedded using the same model, and the closest vectors from the VDB are retrieved using an approximate or exact nearest-neighbor algorithm [10]. The corresponding text is then appended as context to a prompt for a synthesizing LLM, which generates the final answer.

Even in this basic setup, there are numerous design choices across modules: embedding models (e.g., see MTEB [11] for SOTA models by task and size), text splitters (e.g., Sentence, Recursive, Token; see [12]), synthesizing LLMs (e.g., gpt-4o, gpt-4o-mini, clause-3.5-sonnet, llama-3.3-70B), retrievers (e.g., sparse [13], dense [1], hybrid [14]), and VDBs (e.g., Pinecone [15], FAISS [16], Neo4j [17]). Each of these often requires tuning hyperparameters for optimal performance. For example: with the Sentence splitter, what are the ideal chunk size and overlap? For the hybrid retriever, how should weights be balanced between sparse and dense retrievals, and how many nearest neighbors (k) should each return?

These choices are interdependent—e.g., large retrieval sizes may necessitate more capable LLMs, affecting latency, accuracy, and cost. Even within this "vanilla" setup, the combinatorial explosion of configurations across modules leads to hundreds of possible RAG flows.

In practice, modern RAG flows—both agentic [18] and non-agentic [19]—are often far more complex, incorporating additional modules such as verifiers [20], rewriters [21], rerankers [22], and iterative reasoning at inference time [23]. While these added components can enhance performance, they also increase cost and latency, without guaranteeing better task accuracy.

This complexity raises several key questions for AI application development: ① How should one choose an appropriate flow for a given application? ② Should the flow be agentic or non-agentic?

③ Which embedding model and LLM should be used, and how should their hyperparameters be tuned? ④ Is the flow's latency acceptable for the use case? ⑤ What is the maximum achievable accuracy within a fixed budget? ⑥ What is the minimum cost to meet a required accuracy? ⑦ Within a flow, which components have the greatest influence on accuracy, latency, and cost?

To address these questions, we introduce `syftr`, a system that efficiently explores an enormous space ( $10^{23}$) of agentic and non-agentic RAG flows to identify a Pareto frontier—an optimal tradeoff curve [24] between task accuracy and cost. To our knowledge, `syftr` is the first system to perform multi-objective search over generative AI flows.

Unlike traditional multi-objective optimization, searching over AI flows is uniquely challenging due to their compositional structure, complex module interactions, and the stochastic and costly nature of LLM components. These factors create a high-dimensional, non-convex search space requiring specialized techniques beyond standard optimization:

- For grounding datasets with labeled QA pairs, `syftr` uses multi-objective Bayesian Optimization [25] to efficiently search both agentic and non-agentic flows.
- A novel early-stopping mechanism halts flow evaluation when further improvement to the Pareto frontier is unlikely.
- Compared to default flows in libraries like LlamaIndex, `syftr` finds Pareto-dominant flows that are on average 6% more accurate at the same baseline cost, conversely 37% cheaper for the same baseline accuracy, across multiple RAG benchmarks.
- `syftr` supports holistic evaluation of modules, flows, embedding models, and LLMs—e.g., assessing a new LLM across diverse datasets and flows.
- When applied to a new grounding corpus, `syftr` can warm-start optimization using prior trials, enabling faster convergence to near-optimal solutions.

## 2  Related Work

The field of AutoML [26] has made significant strides in automating the construction of machine learning pipelines by searching over large combinatorial spaces of preprocessing, modeling, and post-processing components [27, 28]. At the core of many AutoML systems lies hyperparameter optimization (HPO), with Bayesian Optimization (BO) playing a particularly prominent role in navigating complex search landscapes [25, 29]. Closely related, Neural Architecture Search (NAS) extends the AutoML paradigm to learn model architectures optimized for both performance and hardware constraints, balancing tradeoffs across accuracy, latency, and throughput [30, 31, 32].

Within the domain of retrieval-augmented generation (RAG), `AutoRAG` [33] is the most closely related system to `syftr`. It employs greedy, module-wise optimization to tune flow components for a single performance objective. In contrast, `syftr` performs global multi-objective search, capturing dependencies between modules and avoiding performance degradation due to local optima [34, 35].

Other recent frameworks take alternative approaches to optimizing or constructing LLM-based pipelines. DSPy [36] offers a declarative interface for authoring modular flows, where individual components can be "compiled" through prompt tuning, though it does not address flow-level search or optimization. Trace [37] and TextGrad [38] reframe flow construction as a program synthesis task, using LLMs themselves as optimizers for non-differentiable programs [39]. `syftr` is complementary to these systems: it provides a principled optimization layer that can evaluate and improve entire flows, regardless of how they are constructed.

DocETL [40] represents a different branch of related work focused on constructing document-centric ETL pipelines. It uses rule-based, agentic search techniques inspired by classical cascade systems [41] to optimize data transformation for accuracy. While DocETL and `syftr` both explore modular pipeline optimization, they differ substantially in both domain and methodology: DocETL

targets static ETL workflows and optimizes for accuracy alone, whereas `syftr` is designed for dynamic RAG flows and jointly optimizes multiple competing objectives using probabilistic search.

In summary, prior work has explored various forms of flow construction, optimization, and compilation, from AutoML and NAS to LLM-driven pipeline synthesis. `syftr` builds on this foundation by introducing a unified, multi-objective Bayesian optimization framework for RAG, enabling efficient discovery of high-performing flows across diverse tradeoff surfaces.

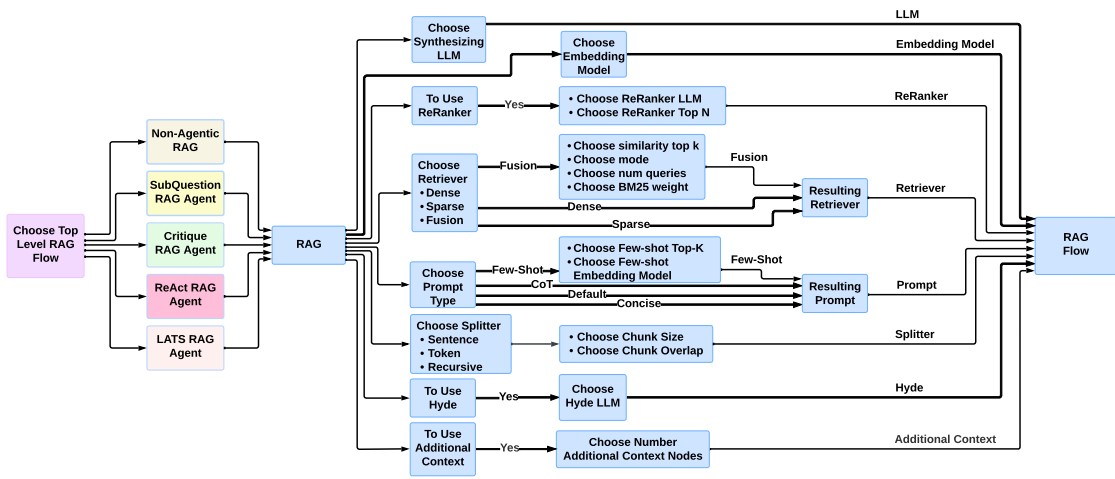

Figure 3: `syftr` RAG hierarchical search space includes 5 top-level flows – 4 agentic and 1 non-agentic with a total of $10^{23}$ unique flows. The agentic flows use the `RAG` flow as a subroutine while adding their own unique hyperparameters.

## 3 Search Space

The `syftr` search space (Fig. 3) consists of five top-level `RAG` flows: four agentic and a single non-agentic (imperative) `RAG` flow. Once a flow is chosen, then further choices have to be made for each of its modules. In turn, each module choice requires more choices to be made for hyperparameters unique to that module. This results in a hierarchical search space containing $10^{23}$ unique flows.

The search space includes four Agentic flows: SubQuestion RAG Agent [42], Critique RAG Agent [43], ReAct RAG Agent [44], and LATS RAG Agent [45]. These flows each have additional hyperparameters (not shown) and use the RAG flow as a tool so their individual search spaces subsumes that of the `RAG` flow. Major components of the search space include ① Synthesizing LLMs: We chose the list of LLMs in Table A2 to cover different prominent frontier model providers, both closed and open weights, and a range of sizes from flash/mini to the largest ones. ② Embedding Models: We selected the top-performing models from the Massive Text Embedding Leaderboard [11], focusing on those optimized for retrieval and with fewer than 500M parameters at the time of writing. Each model is served via the autoscalable HuggingFace Dedicated Inference Service [46]. ③ Other Modules: Components such as `ReRanker`, `Retriever`, `Splitter`, and `HyDE` are widely used in modern RAG implementations. For detailed descriptions of these modules see Appendix A1.

This search space is challenging not only because of its large size, but also because evaluating each candidate is computationally expensive. Each evaluation involves constructing the full flow and running it on an evaluation dataset, making function evaluation costly in both time and resources. Therefore, to make search as efficient as possible we leverage recent advances in multi-objective Bayesian Optimization (BO) optimized for hierarchical search spaces [47, 48].

## 4 Multi-Objective Bayesian Optimization

The *Pareto-frontier* (or *Pareto-front*) represents the set of non-dominated solutions in a multi-objective optimization problem. A solution is Pareto-optimal if no objective can be improved without worsening at least one other objective [49]. Our aim is to efficiently search the hierarchical space to identify a set of Pareto-optimal flows that tradeoff between multiple objectives. To do so, we require an optimization method well suited for both hierarchical search and multi-objective optimization. Multi-Objective Tree-of-Parzen Estimators (MO-TPE) [47, 48] satisfies these criteria.

The Tree-Structured Parzen Estimator (TPE) [50] introduced a novel approach to Bayesian Optimization (BO). Instead of modeling the objective function directly, TPE models two conditional densities: 1. $l(x)$ likelihood of hyperparameters $x$ leading to good performance and 2. $g(x)$ likelihood of hyperparameters $x$ leading to poor performance. By dividing the observed data into good and bad subsets based on a quantile threshold (e.g, top 25%) TPE transforms the optimization into a density estimation problem. The method leverages tree-structured distributions to represent hyperparameter spaces, allowing for better handling of complex and hierarchical search spaces (also termed conditional hyperparameters in the literature).

Ozaki et al. [47] extended TPE to the multi-objective setting (MO-TPE) by approximating the Pareto-frontier (a set of tradeoff solutions where no objective can be improved without worsening another). MO-TPE uses a weighted linear scalarization approach to convert multiple objectives into a single scalar objective for evaluation and an expected hypervolume improvement (EHVI)-based strategy to guide the search. EHVI measures the contribution of a new sample to the hypervolume dominated by the Pareto-frontier, ensuring exploration of the most promising regions. `syftr` uses Optuna's [51] MO-TPE implementation.

Once a candidate flow is constructed, it is evaluated on a set of evaluation question-answer pairs. Evaluation sets often consist of thousands of question-answer pairs. This is time-consuming and expensive, as calls are made to the embedding model, synthesizing LLM, and one or more judge LLMs. We introduce `Pareto-Pruner`, a novel pruning technique that estimates the confidence intervals around both task accuracy and cost and *early stops* evaluations once the confidence interval bounding box drops below the current Pareto-frontier.

Specifically, during the evaluation phase of a trial, intermediate results are sent to the pruner. The `Pareto-Pruner` evaluates whether the current point has a reasonable chance to improve the current estimate of the Pareto-frontier if evaluation is continued. `Pareto-Pruner` computes the upper-left corner $p$ of the confidence interval bounding box shown in Fig. 4. If this point is above the current Parto-frontier, then the evaluation continues. Conversely, if $p$ falls below the current Pareto-frontier, the trial already dominated by other Pareto-optimal solutions, and evaluation resources can be safely spent elsewhere.

We model the uncertainty in accuracy and cost dimensions independently of each other: costs are a right-tailed distribution that is bounded to the left by zero, while accuracy is bounded in [0, 1] and is more centered, with some top-performing, some low-performing, and a lot of mediocre flows. For a given dataset, to model costs, we fit a log-normal distribution and use the 90% confidence interval ($z = 1.645$). For accuracy, we fit a normal distribution and use a confidence level of 90%. Appendix A2 visualizes the accuracy and cost distributions fit to each of the datasets. Experiments to assess the efficacy of the `Pareto-Pruner` may be found in Appendix A3.

**Seeding the Optimizer**: Given the sheer size of the search space, it is important to help the optimizer bootstrap with informative priors via a seeding procedure. The goal of custom seeding is two-fold: 1. To ensure that specific flows that are commonly used by the community are definitely part of the search space that is evaluated and 2. To inject any domain-specific knowledge of the search space so that the MO-TPE starts with a useful prior. `syftr` supports several types of *custom seeding* routines which run before the optimization process starts: *Static Seeding*: `syftr` uses a set of commonly-used "standard" flow configurations, obtained by systematically iterating over

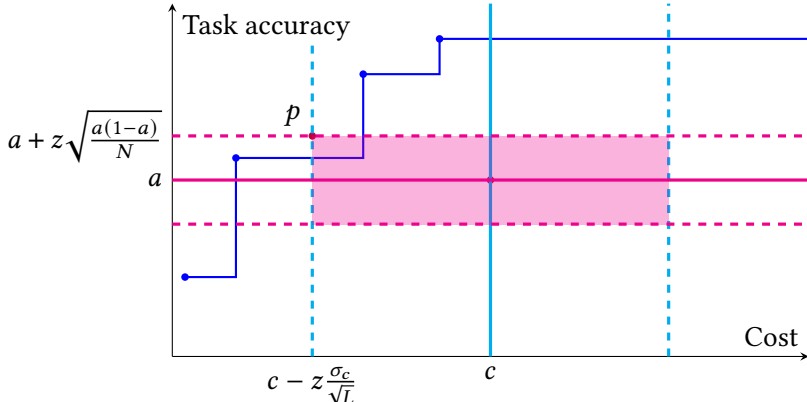

Figure 4: `Pareto-Pruner` estimates confidence intervals around task accuracy and cost for a given flow, and will early-terminate flows whose upper-left confidence point ($p$) falls below the current Pareto-frontier. $c$ is the P80 cost, $a$ is the average accuracy, $L$ is the number of successful evaluations, $N$ is the number of total evaluations including errors caused by issues like content filtering, endpoint rate limit hits and agentic flow failures due to improper tool usage. $\sigma_c$ is the standard deviation of the costs of the current evaluation, and $z$ is the standard score for a normal distribution and controls the sensitivity of the `Pareto-Pruner`.

key parameters such as the synthesizing LLM and embedding model. This ensures that the initial search includes well-established baseline configurations. A complete list of these standard flows is provided in Table A5. *Random Seeding*: flows are generated by randomly sampling from the search space. *Transfer Seeding*: flows from a previous search can be used to jump-start the optimization process as shown in Appendix A5. Unless otherwise specified, experiments in this paper use a mix of random and static seeding before starting optimization.

## 5    Datasets and Evaluation Protocol

**Datasets**: We evaluate `syftr` across a set of RAG benchmarks, each partitioned into `train`, `test`, and `holdout` sets. Flow optimization is always performed against the `test` partition, while the `train` set may be used for dynamic prompting (via the Dynamic Few-Shot Retriever) and, in the future, for LLM fine-tuning. The `holdout` set is reserved for final evaluation. Some datasets also include a small `sample` partition for development. Each dataset includes a grounding corpus e.g., PDFs, HTML, or plain text which is used during retrieval.

Our benchmark suite includes HotpotQA [52], a multi-hop QA dataset from Wikipedia; FinanceBench [53], a challenging financial QA dataset over SEC filings; CRAG [54], a broad RAG benchmark derived from web data with questions spanning five topics; InfiniteBench [55], a long-context reasoning dataset based on altered public domain books; and DRDocs, a synthetic QA dataset built from DataRobot's documentation. For preprocessing, we convert HTML and PDF content to Markdown (e.g., using Aryn DocParse [56] and html2text [57]). Each dataset presents different challenges—such as multi-hop reasoning, numeric computation, or retrieval over long contexts—enabling a robust evaluation of `syftr` across domains, model sizes, and flow configurations. Dataset details are located in Appendix A6.

**Evaluation Protocol**: We use LLM-as-a-Judge [58] to evaluate flow-generated answers against ground-truth dataset answers. Given known sensitivities of judge models to prompts, answer formatting, and model bias [59, 60], we conducted a study comparing various evaluator configurations against human judgment on 447 responses generated by different flows across multiple datasets. Based on this study, we selected the Random LLM evaluator configuration, which randomly selects

a judge LLM for each evaluation, providing diverse assessments at reasonable cost. Details of this study may be found in Appendix A7.

# 6   Results

We present several empirical studies demonstrating syftr's capability to identify Pareto-optimal solutions tailored to specific operational constraints.

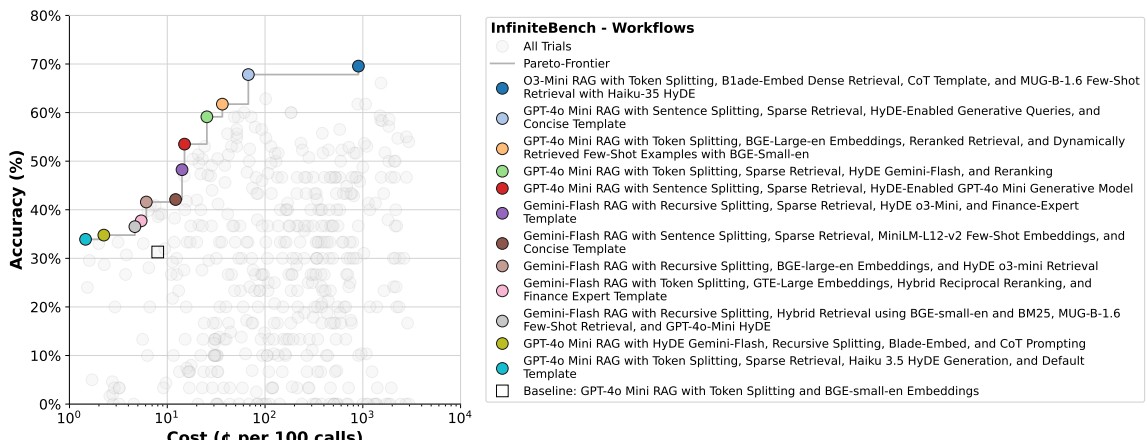

Figure 5: Multi-Dataset Study: Pareto-frontier for InfiniteBench; See Appendix A8 for other datasets. Colored dots represent flows whose key components are described in the legend. Legend items are sorted by descending accuracy, except for the flow denoted *Baseline* at the bottom, which is a baseline RAG flow that is similar to LlamaIndex default settings and uses gpt-4o-mini as the synthesizing LLM and bge-small-en-v1.5 embeddings. Note the x-axis is log scale.

**Multi-Dataset Study**: We run syftr on a space of relatively smaller LLMs in Table A2 containing both agentic and non-agentic flows to find Pareto-frontier between accuracy and cost. Fig. 5 shows the Pareto-frontiers found for all datasets described in Section 5. Key observations: ① Non-agentic RAG flows appear on the Pareto-frontier far more frequently than agentic RAG flows. Non-agentic RAG flows are cheaper and faster to run, leading the optimizer to focus its exploration in this area. ② GPT-4o-mini shows up frequently in Pareto-optimal flows on all of the datasets, indicating its quality as a synthesizing LLM. ③ RAG enhancements such as HyDE and Reranking are occasionally Pareto-optimal, indicating they are situationally beneficial. ④ Reasoning models such as o3-mini provide superior performance on FinanceBench. We conjecture that due to the quantitative nature of the dataset and multi-hop reasoning required to answer questions, GPT-o3-mini does well here. ⑤ All datasets show Pareto-frontiers that flatten out: *large increases in cost bring diminishing returns in accuracy after initial steep rise.* We observe that often marginal accuracy is to be gained from orders of magnitude increase in cost. This allows practitioners to pick appropriate points of operation. ⑥ Compared to the default RAG flow in LlamaIndex, (at the time of writing: gpt-4o-mini as the synthesizing LLM and bge-small-en-v1.5 as embedding model), syftr consistently finds flows which are 6% more accurate while retaining identical costs, or conversely decrease costs by 37% while retaining identical accuracy. (See Fig. A9 for details).

**Agentic Study** This study explores the space of agentic flows on the challenging FinanceBench dataset. After evaluating 476 trials, we identified flows achieving over 70% accuracy on this difficult benchmark. Notably, these high-performing solutions have significantly higher costs, with the most accurate flows approaching $10 per call, and the total cost for this study surpassing $2000. Therefore, in scenarios demanding high accuracy on complex tasks, agentic flows can deliver high performance at the expense of increased cost and latency. When comparing different agentic flows,

the SubQuestion Agent dominates the high-cost, high-accuracy region, whereas the ReAct and Critique RAG Agents offer Pareto-optimal solutions in lower-cost regimes.

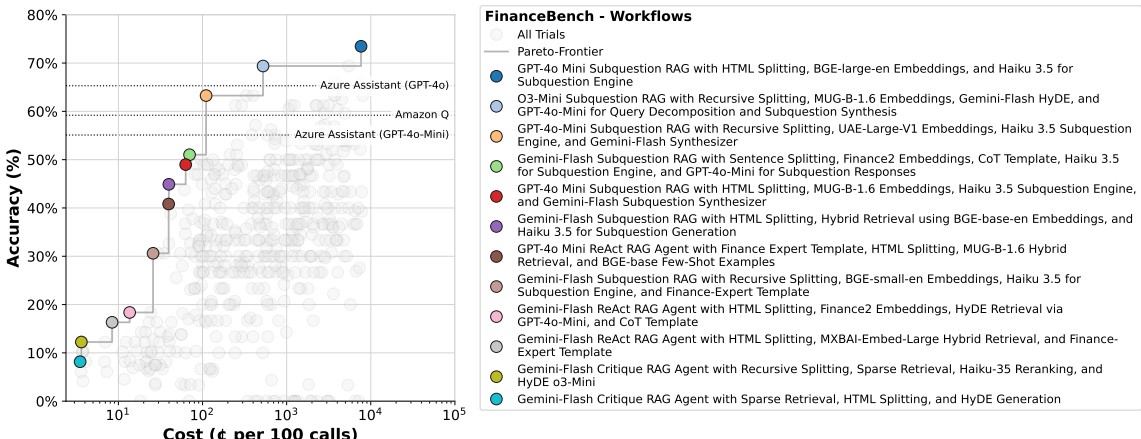

Figure 6: Agentic Study: Agents deliver high performance at the expense of additional cost and latency.

**Large Models**: To evaluate the potential benefits of larger LLMs, we selected Pareto-optimal flows from the CRAG3 music dataset and upgraded their associated LLMs: `GPT-4o-mini` to `GPT-4o`, `Gemini-Flash` to `Gemini-Pro`, and `Anthropic-haiku-35` to `Anthropic-sonnet-35`. These upgrades were applied across all LLM-based components, such as synthesizing LLM and HyDE.

Table 1: Effects of upgrading LLM Size on Pareto-optimal CRAG3 music flows. On average, accuracy increases by 17.3 percentage points and cost increases by a factor of 67.

| Type | Accuracy (%) | | Cost ($/call) | | Accuracy Delta (pp) | Cost Multiplier |
|---|---|---|---|---|---|---|
| | Small | Large | Small | Large | | |
| RAG | 55.8 | 67.6 | 0.000197 | 0.003214 | 11.7 | 16.3 |
| RAG | 67.6 | 73.5 | 0.000392 | 0.006585 | 5.8 | 16.7 |
| RAG | 70.5 | 85.2 | 0.000740 | 0.012132 | 14.7 | 16.3 |
| RAG | 82.3 | 85.2 | 0.007168 | 0.119495 | 2.9 | 16.6 |
| Critique RAG Agent | 50.0 | 90.6 | 0.000118 | 0.029929 | 40.6 | 253.6 |
| ReAct RAG Agent | 41.1 | 71.4 | 0.000099 | 0.000506 | 30.2 | 5.1 |
| ReAct RAG Agent | 58.8 | 73.5 | 0.000257 | 0.050662 | 14.7 | 197.1 |
| RAG | 73.5 | 91.1 | 0.004549 | 0.073003 | 17.6 | 16.0 |

As shown in Table 1, upgrading to larger LLMs consistently leads to significant accuracy improvements, averaging 17.3%. However, this comes with a steep 67-fold increase in cost which is particularly pronounced for agentic flows that involve multiple LLM calls per execution. These results highlight the critical trade-off practitioners face when selecting more powerful LLMs: meaningful accuracy improvements must be weighed against substantial operational cost increases. Fig. 7 shows as the models are swapped with their larger variants how the Pareto-frontier shifts. There are accuracy increases across the board but many flows on the original Pareto-frontier fare no longer on the new Pareto-frontier showing that local optimizations alone are often not Pareto-optimal.

**Latency Optimization**: `syftr` is capable of optimizing over a variety of objectives. To demonstrate, we perform a multi-objective optimization over accuracy and latency objectives on the FinanceBench dataset. Low latency is important for real time tasks such as RAG systems that would engage in

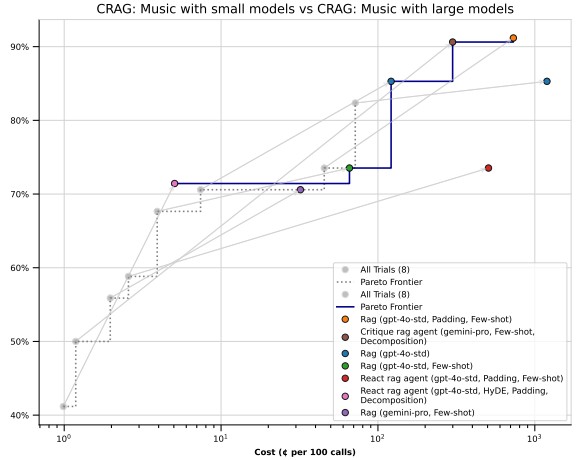

Figure 7: Effects of upgrading LLM size on CRAG3 music: The original Pareto-frontier shown in gray is shifted up and to the right, reflecting the increased accuracy and cost. Even though accuracy increases across the board, not all flows on the original Pareto-frontier remain on the new Pareto-frontier, indicating that component-wise optimizations are not always Pareto-optimal.

live conversations with a user. Fig. 8 shows only the lowest-accuracy flows could achieve latencies required for real time voice conversation (typically <400ms), with highest accuracy flows having latencies of 30 seconds or more. A sweet spot emerges around 10 seconds of latency, where highly accurate flows are already available. By having the entire Pareto-frontier of solutions, it's possible to quickly assess these tradeoffs and select a flow that fits requirements.

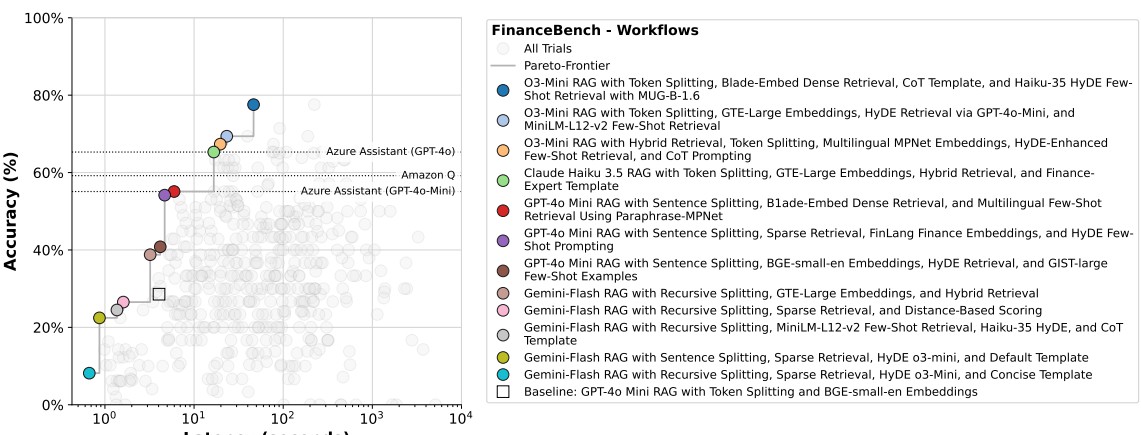

Figure 8: Latency optimization for FinanceBench. Flows with Gemini-Flash as a response synthesizer are fast but also have low accuracy on FinanceBench. Flows with the reasoning model O3-Mini as synthesizer show the best accuracy for this challenging dataset. Flows with GPT-4o-Mini provide a good tradeoff between speed and accuracy. The horizontal lines are the accuracies obtained by commercial third-party proprietary RAG solutions such as Azure Assistant and Amazon Q. At the time of writing the accuracies obtained significantly lag that of the best accuracies found by `syftr`.

**Third-Party Comparisons**: Amazon Q [61] uses models from Amazon Bedrock, primarily Amazon Titan and GPT-based generative AI. Azure Assistant [62], built on Azure OpenAI Service, enables developers to create goal-directed agents with persistent memory, tools, and function calling. We evaluated both using their default configurations on FinanceBench: `syftr` achieves 77.6% accuracy (Fig. 8), outperforming Azure Assistant (65.3%) and Amazon Q (59.2%). This gap is expected—third-party assistants are built for broad use, while `syftr` is tuned to the dataset. Still, it highlights a key insight: custom flows can significantly outperform off-the-shelf solutions!

# 7   Hardware and Costs

What are the costs associated with running an optimization? How should practitioners evaluate whether to initiate an optimization, given uncertainty about whether it will yield a better-performing flow? How long should an optimization be run? And what kind of hardware is required to execute `syftr` effectively?

All experiments in this paper were conducted on a compute cluster equipped with 696 CPUs, 4 Nvidia A100 GPUs, and 4 Nvidia A6000 GPUs. The GPUs were primarily utilized for efficient embedding computations. See Section A1.5 for additional implementation and scaling details.

Table 2 summarizes the computational outcomes and costs for the main studies. Each study aimed for approximately 500 successful trials to ensure broad exploration of the search space across varied datasets and flow types. However, execution encountered real-world challenges such as LLM API rate limits, transient endpoint failures, and edge cases in the parameter space, which contributed to the observed number of failed trials.

The cost of each study varied significantly—from \$125 to over \$2300—depending on the configuration and complexity of the flow being optimized. Assessing the cost-effectiveness of `syftr` requires context-specific considerations, including the application domain, dataset size, search configuration, and available baselines. Given this variability, a general-purpose cost-effectiveness analysis is beyond the scope of this paper. Nonetheless, our results show that `syftr` can consistently discover high-performing flows across diverse tasks at a reasonable cost, underscoring its practical utility for real-world deployment.

Table 2: Runtime and costs for the main studies.

| Study Name | Successful | Failed | Pruned | Trials/hr | Time/Trial | Costs (\$) |
|---|---|---|---|---|---|---|
| FinanceBench - Agents | 549 | 5 | 60 | 33 | 1h07m | 2310.30 |
| InfiniteBench - Agents | 406 | 8 | 87 | 87 | 0h32m | 887.29 |
| CRAG: Music | 502 | 72 | 47 | 12 | 1h35m | 380.91 |
| CRAG: Sports | 390 | 61 | 136 | 11 | 1h44m | 629.67 |
| DR Docs | 525 | 30 | 69 | 33 | 0h58m | 339.66 |
| FinanceBench | 506 | 76 | 84 | 19 | 1h13m | 386.27 |
| HotpotQA | 503 | 17 | 204 | 33 | 0h50m | 125.13 |
| InfiniteBench | 475 | 4 | 52 | 53 | 0h26m | 764.99 |
| FinanceBench - Latency | 478 | 13 | 44 | 56 | 0h48m | 1359.08 |

# 8   Discussion and Future Work

We introduced `syftr`, a system that efficiently explores a vast search space of $10^{23}$ flows to identify Pareto-optimal solutions balancing accuracy and cost. Our results show that generative AI pipeline performance depends heavily on dataset characteristics. Rather than replacing human expertise, `syftr` empowers data scientists and engineers with a data-driven tool to rapidly navigate complex design decisions and optimize performance on new datasets.

We are expanding `syftr` to support multi-agent workflows [63, 64], and plan to integrate prompt optimization into the Bayesian search loop, given its major impact on performance [37, 36]. We believe tools like `syftr` will be essential for navigating the generative AI design space, enabling tailored, cost-effective, and accurate solutions.

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

## A1 Search Space Details

Table A1: `syftr` choices for each module and their search spaces for the `RAG` workflow. For discrete search spaces we use the convention [start:step:stop]. For example, [2:1:20] means the set $\{2, 3, 4, \ldots 20\}$. logspace[start:stop] means the interval is sampled in log space. E.g., logspace[1:128] uses a log-uniform distribution over the interval which results in smaller values are sampled more finely than larger values. The column **Module** lists the modules, **Module Space** lists the choices for the module along with choice-specific hyperparameters. In the **Shared Space** column the hyperparameters which apply to all choices for a particular module are listed for brevity. For e.g., the choice of `chunk-size` applies to all choices for the Splitter module.

| Flow | Module | Module Space | Shared |
|------|--------|--------------|--------|
| RAG | Synthesizing LLM | LLMs in Table A2 | |
| | Reranker | LLMs in Table A2
top-k ([2:1:20]) | |
| | Embedding Model | Models in Table A3 | |
| | Splitter | Recursive
Token
Sentence | chunk-size
(logspace[256:4096])
chunk-overlap
([0.0:0.1:0.7]) |
| | HyDE | LLMs in Table A2 | |
| | Retriever | Dense
Sparse
Fusion
—num-queries [1:1:20]
—mode (4)
—bm25-weight [0.1:0.1:0.9] | top-k ([2:1:20]) |
| | Prompt | Default
Concise
CoT | |
| | Dynamic Few-Shot Retriever | top-k ([2:1:20])
Embedding Model Models in Table A3 | |
| | Additional Context | —num-nodes (logspace[2:20]) | |

Tables A2 and A3 list the LLMs and embeddings models used in `syftr` search space. We provide a brief overview of the function served by various module choices in the search space for completeness.

**ReRanker**: Rerankers [65, 66] are optionally used in RAG pipelines to improve the quality of retrieved information before it is added as additonal context to a synthesizing LLM. Rerankers evaluate and reorder candidate chunks retrieved by an initial retriever. These models can leverage contextual features, query-document relationships, or semantic understanding to assign scores to documents, prioritizing those most relevant to the query. But at the same time reranking adds additional complexity and increasingly synthesizing LLMs are increasing in base capability where this may not be needed. By incorporating the use of rerankers into the search space, `syftr` surfaces

| Small LLMs | Embedding Model Name |
|---|---|
| anthropic-claude-haiku-3.5@20241022 | bge-small-en-v1.5 |
| o3-mini@2024-09-12 | bge-large-en-v1.5 |
| gpt-4o-mini@2024-07-18 | gte-large |
| gemini-flash-1.5-002 | mxbai-embed-large-v1 |
| **Large LLMs** | UAE-Large-V1 |
| anthropic-claude-sonnet-3.5-v2@20241022 | GIST-large-Embedding-v0 |
| mistral-large-2411 | b1ade-embed |
| llama-3.3-70B-instruct | MUG-B-1.6 |
| gemini-pro-1.5-002 | all-MiniLM-L12-v2 |
| gpt-4o@2024-11-20 | paraphrase-multilingual-mpnet-base-v2 |
| | bge-base-en-v1.5 |

Table A2: List of small and large LLMs          Table A3: List of Embedding models

flows where the use of rerankers is automatically decided. The list of LLMs in Table A2 are searched over for use as rerankers.

## A1.1    Retrievers

syftr searches over three kinds of retrievers 1. Dense 2. Sparse and 3. Fusion retrievers.

**Dense retrievers** represent queries and documents as dense vectors in a continuous vector space, learned through neural embeddings. Techniques such as BERT-based models and contrastive learning are often employed to optimize embeddings such that similar queries and documents are mapped to proximate regions in the vector space. These retrievers leverage semantic matching, enabling them to handle complex natural language queries effectively. Dense retrievers are especially advantageous when dealing with semantic nuances but can be computationally intensive due to the need for vector indexing and similarity computation in high-dimensional spaces. In syftr computing vector embeddings of queries and grounding datasets for dense retrieval is one of the most expensive steps. The list of embedding models in Table A3 is searched over for creating VDBs from grounding datasets and to embed the query during retrieval.

**Sparse retrievers**, like the widely used BM25 [13], represent documents and queries as sparse vectors in a high- dimensional space. These models rely on keyword-based matching, prioritizing exact term overlaps between queries and documents. Sparse retrievers are computationally efficient and interpretable but may struggle with capturing semantic relationships between words, such as synonyms. They excel in scenarios where precise keyword matches are crucial.

**Fusion retrievers** [14] combine the best of both worlds using both dense and sparse retrievers and fusing the retrieval results using various schemes before presenting the combined context to the synthesizing LLM. If the Fusion retriever is chosen, additionally hyperparameters specific to it like top-k, mode and num queries have to be additionally sampled. The Fusion retriever breaks down the original query into num queries and does retrieval with each query. It then fuses the retrieved chunks of text across all queries using reciprocal rank scores.

## A1.2    Prompt Strategies

syftr considers four discrete prompt strategies in its search space: Default, Concise, CoT and Dynamic Few-Shot. See Table A4 for the exact prompt templates. The Default prompt is a generic prompt while the Concise prompt asks the LLM to be succinct while answering. The CoT [67]

Table A4: Prompt templates used in the search space of `syftr`. `query_str` is filled with the actual query. `few_shot_examples` is dynamically filled with example query-answer pairs from a predefined pool based on similarity to the query.

| Variant | Description |
| --- | --- |
| default | You are a helpful assistant. Answer the provided question given the context information and not prior knowledge. Question: {query_str} Answer: |
| concise | You are a helpful assistant. Answer the provided question given the context information and not prior knowledge. Be concise! Question: {query_str} Answer: |
| CoT | You are a helpful assistant. Answer the provided question given the context step-by-step. Question: {query_str} Answer: |
| dynamic-few-shot | You are a helpful assistant. Answer the provided question given the context information and not prior knowledge. Some examples are given below. {few_shot_examples} Question: {query_str} Answer: |

prompt uses the widely used chain-of-thought prompting technique to encourage the LLM to step through a reasoning process before answering.

In `Dynamic Few-Shot` prompting [68], instead of using a static set of examples, examples are selected based on the query to dynamically select example question-answer pairs to construct a prompt. The examples are selected from a pre-defined pool based on similarity to the query. The idea is that since LLMs are good few-shot in-context learners [69], demonstrating how to do similar tasks in the prompt itself results in better performance.

## A1.3 Text Splitters

`syftr` searches over `Recursive`, `Sentence` and `Token` text splitters in its search space. For each splitter two hyperparaters (`chunk-size` and `chunk-overlap` are also searched over.

**Recursive** splitter decomposes text iteratively by breaking it down into smaller and smaller components, starting from larger linguistic units (e.g., paragraphs) and working toward finer granularity (e.g., words). Token splitter segment text into tokens, which are the smallest meaningful units, such as words, punctuation marks, or symbols.

**Sentence** splitter identifies and separates tokens into individual sentences, typically by detecting punctuation markers (e.g., periods, question marks) and leveraging language-specific rules to handle edge cases (e.g., abbreviations). We utilize the implementations of these splitters in the LlamaIndex library [9].

## A1.4 HyDE

Instead of directly retrieving the most relevant chunks from a VDB, `HyDE` [21] instead zero-shot instructs a LLM to generate a hypothetical document. This document may contain false details. A contrastively trained encoder encodes the document into an embedding vector. This embedding vector of a hypothetical document is then used to search for real documents in the VDB. `HyDE` adds significant complexity and latency to a RAG pipeline but often helps improve accuracy. But by being part of the search space, `syftr` automatically helps decide if the additional complexity is worth it.

### A1.5 Implementation Details and Infrastructure Challenges

**Compute infrastructure**: `syftr` requires significant infrastructure to scale up the search process. One source of complication are the heterogeneous requirements needed for constructing different flows. For example, if a flow uses a large embedding model, it will require a larger GPU with more memory than a flow which utilizes a smaller embedding model. This step has to be repeated many times for different flows. We build on top of Ray [70], leveraging its Ray Tune [71] integration with Optuna [51]. Ray provides an easy-to-use scalable Pythonic language to scale up jobs across a heterogeneous cluster of nodes with Nvidia GPUs.

For computing embeddings on large datasets with different embedding models, we utilize HuggingFace Dedicated Inference endpoints [46]. This allows us to put smaller models such as `BAAI/bge-small-en-v1.5` on Nvidia T4 GPUs, while larger models such as `BAAI/bge-large-en-v1.5` are put on L4 GPUs and so on. Each endpoint is set to autoscale to 5 replicas in our experiments.

**Synthesizing LLMs**: `syftr` utilizes the LLMs listed in Table A2, sourced from various API endpoint providers such as Azure and Google Cloud Provider (GCP). As the search progresses, traffic distribution across LLM endpoints varies dynamically. When the optimizer identifies effective flows using a particular LLM, it prioritizes further exploration around that flow, leading to increased demand on the corresponding LLM endpoint. This results in spiky, high-traffic usage of specific models. To manage this variability, we leverage serverless elastic hosting from Azure and GCP, enabling on-demand scaling based on usage. Despite this, large-scale concurrent trials can generate hundreds of requests per minute and process millions of tokens, necessitating robust retry handling to maintain stability. To accommodate different service-level agreements (SLAs) across model providers, `syftr` is designed with a modular architecture, making it easy to integrate and manage endpoints from diverse providers.

**Scaling Challenges**: While building `syftr` we faced significant challenges since this is quite a novel system. We highlight a few of them below:

- As mentioned in Section A1.5, each parallel trial can require different amounts of resources. For example if a trial is tasked with constructing and executing a flow that uses a large embedding model then it will require a bigger GPU and/or a larger fraction of a GPU or even multiple GPUs for efficient embedding computation. This presents a significant challenge even with mature distributed computing frameworks like Ray [70]. Our cluster has relatively more CPUs (696) compared to GPUs (4 Nvidia A100 and 4 A6000 GPUs). We had to manually tune the number of GPUs and CPUs we allocated to each trial to maximize cluster utilization, experimented with CPU ONNX [72] backends for embedding models which were unfortunately $\sim 50\times$ slower than GPUs. If we used only GPUs for embedding models then jobs would get bottlenecked on the relatively small number of GPUs. We solved this issue by utilizing HuggingFace Dedicated Inference Endpoints [46] which allowed us to pick autoscalable GPU inference endpoints with different Nvidia GPUs per embedding model based on model size.
- We also put in a robust timeout system on top of Ray which allowed us to free-up resources from trials which were taking too long or in stalled state.
- In order to distribute datasets across machines in the Ray cluster we use the HuggingFace DataSets library, and an AWS S3 bucket. To minimize data transfer costs we are investigating a robust thread-safe file caching system.
- Due to different endpoint providers having different rate limits and quotas, we had to manage encountering these limits using retry logic with exponential backoff with randomization.
- Different endpoint providers also have differing content filter implementations. We turn off as many of these as we can to minimize losing requests during evaluation.

## A2 Pareto-Pruner Distributions

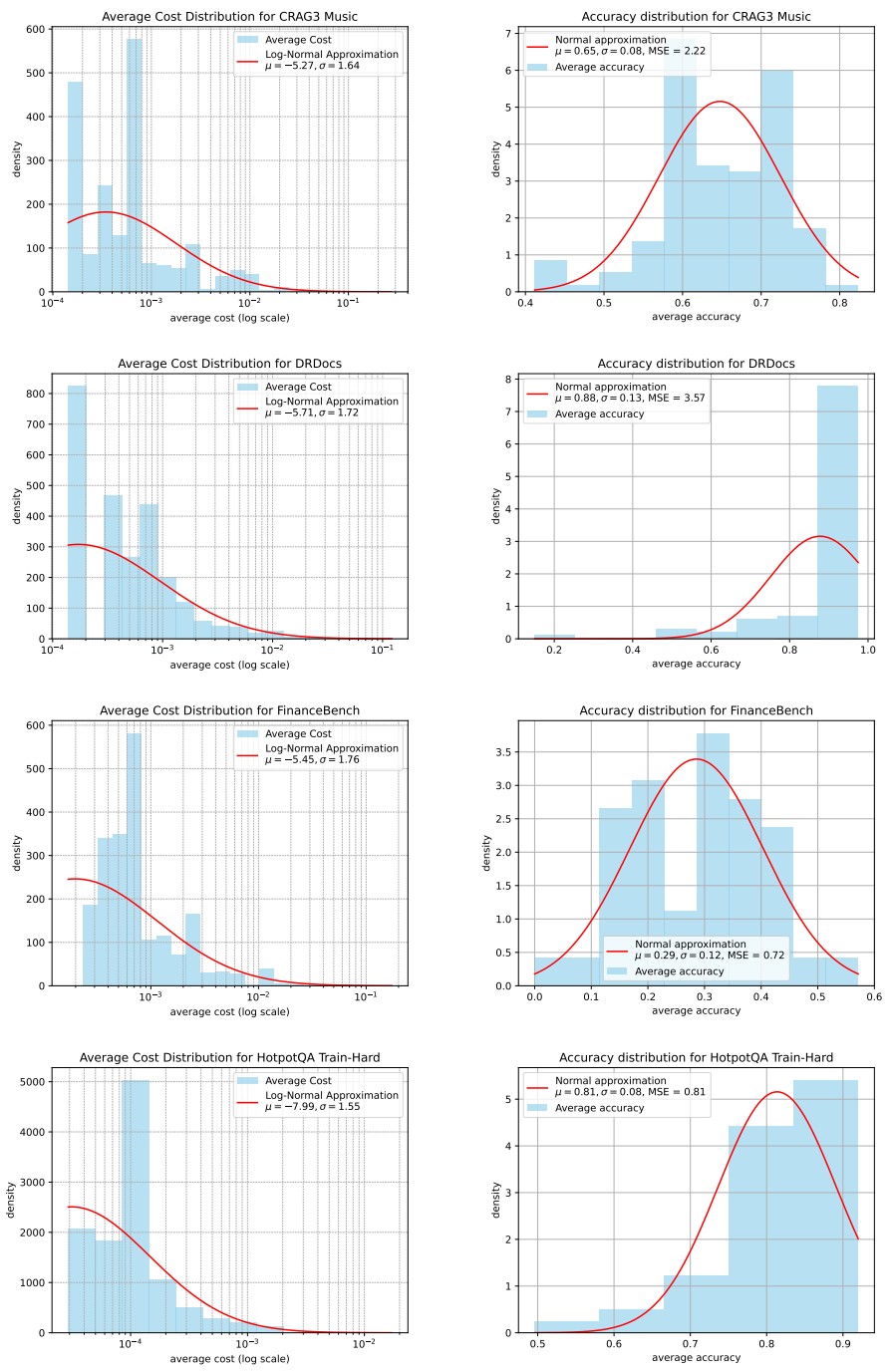

Figure A1: Cost and accuracy distributions used by the Pareto-Pruner for various datasets.

## A3 Pareto-Pruner Ablation

We perform an ablation study to analyze the Pareto-Pruner (Section 4). The objective of this study is to understand the extent to which the Pareto-Pruner saves cost and how it affects the optimization process. Fig. A2 shows that for both datasets explored, the Pareto-Pruner was effective at saving cost and speeding up exploration of the search space.

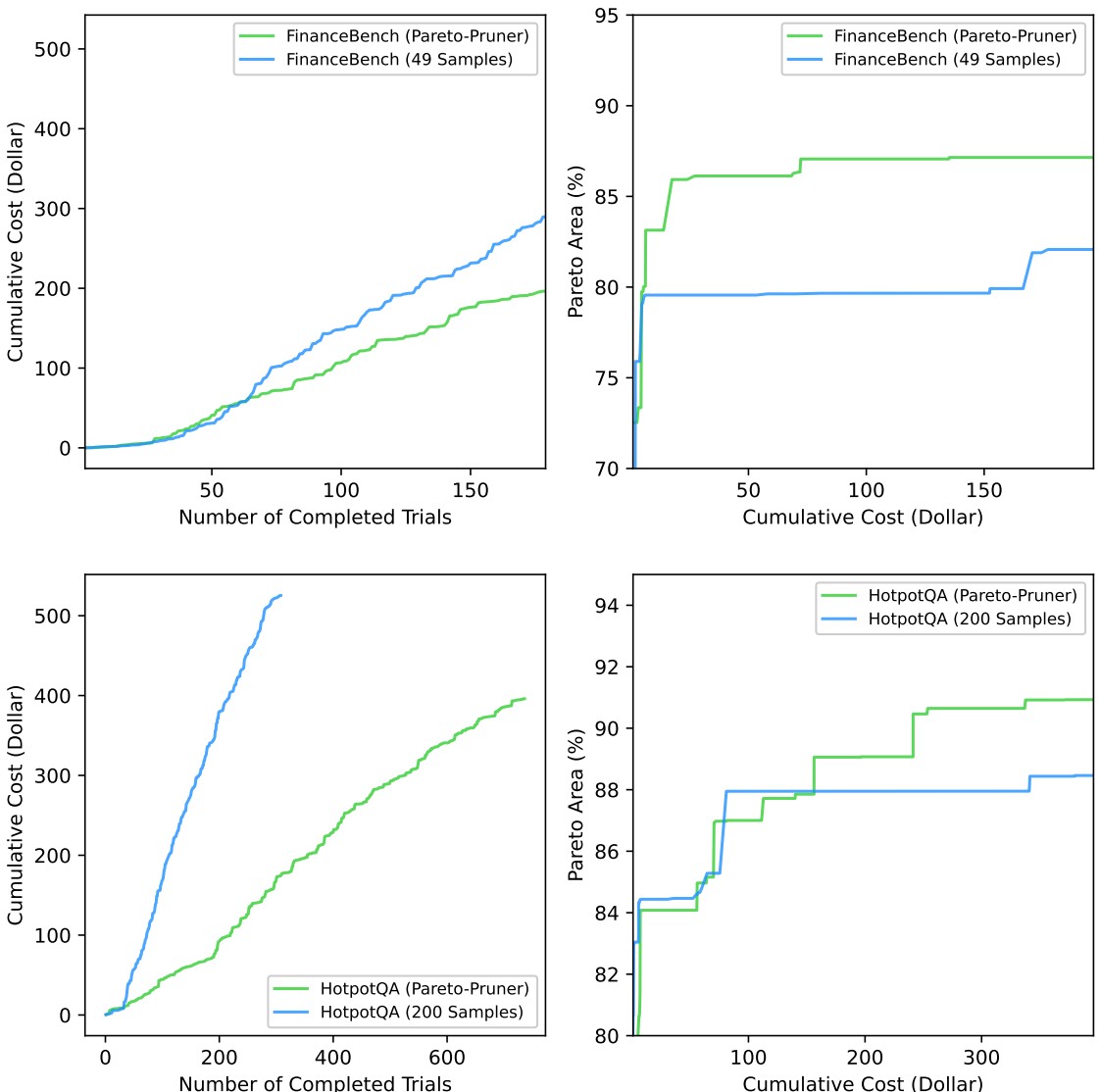

Figure A2: Top-left: the Pareto-Pruner on the FinanceBench dataset reaches a fixed number of trials for a lower cost (green) compared to a no-Pareto-Pruner baseline (blue) which performs all 49 evaluations every trial. Top-right: for a fixed budget, the Pareto-Pruner (green) generally produces a Pareto-curve that covers more of the Pareto-area than the baseline (blue). Second row: same for the HotpotQA dataset that features 200 evaluations per trial for the baseline, allowing Pareto-Pruner to complete a 300-trial study at a fraction of the cost required without it.

## A4  Seeding the Optimizer

To initialize Bayesian optimization, we seed a trial with static flows, randomly selected flows, flows from past optimizations via transfer learning, or a combination thereof. Table A5 enumerates the standard flows used in static seeding.

Table A5: Static Seeding Flows: Selected parameters of static seeding flows for a RAG-optimization.

| LLM | RAG mode | RAG method | Template | Splitter | RAG embedding | Few-shot embedding |
|---|---|---|---|---|---|---|
| gpt-4o-mini | rag | dense | default | token | BAAI/bge-small-en-v1.5 | |
| gpt-4o-mini | rag | dense | default | sentence | BAAI/bge-small-en-v1.5 | |
| gpt-4o-mini | rag | sparse | default | sentence | | |
| gpt-4o-std | rag | dense | default | token | BAAI/bge-small-en-v1.5 | |
| gpt-4o-std | rag | dense | default | sentence | BAAI/bge-small-en-v1.5 | |
| gpt-4o-std | rag | sparse | default | sentence | | |
| gpt-35-turbo | rag | sparse | default | sentence | | |
| anthropic-sonnet-35 | rag | sparse | default | sentence | | |
| anthropic-haiku-35 | rag | sparse | default | sentence | | |
| llama-33-70B | rag | sparse | default | sentence | | |
| gemini-pro | rag | sparse | default | sentence | | |
| gemini-flash | rag | sparse | default | sentence | | |
| mistral-large | rag | sparse | default | sentence | | |
| gpt-4o-mini | rag | dense | few-shot | sentence | BAAI/bge-small-en-v1.5 | all-MiniLM-L12-v2 |
| gpt-4o-std | rag | dense | few-shot | sentence | BAAI/bge-small-en-v1.5 | all-MiniLM-L12-v2 |
| gpt-35-turbo | rag | dense | few-shot | sentence | BAAI/bge-small-en-v1.5 | all-MiniLM-L12-v2 |
| anthropic-sonnet-35 | rag | dense | few-shot | sentence | BAAI/bge-small-en-v1.5 | all-MiniLM-L12-v2 |
| anthropic-haiku-35 | rag | dense | few-shot | sentence | BAAI/bge-small-en-v1.5 | all-MiniLM-L12-v2 |
| llama-33-70B | rag | dense | few-shot | sentence | BAAI/bge-small-en-v1.5 | all-MiniLM-L12-v2 |
| gemini-pro | rag | dense | few-shot | sentence | BAAI/bge-small-en-v1.5 | all-MiniLM-L12-v2 |
| gemini-flash | rag | dense | few-shot | sentence | BAAI/bge-small-en-v1.5 | all-MiniLM-L12-v2 |
| mistral-large | rag | dense | few-shot | sentence | BAAI/bge-small-en-v1.5 | all-MiniLM-L12-v2 |
| gpt-4o-mini | react-rag-agent | dense | default | sentence | BAAI/bge-small-en-v1.5 | |
| gpt-4o-mini | critique-rag-agent | dense | default | sentence | BAAI/bge-small-en-v1.5 | |
| gpt-4o-mini | sub-question-rag | dense | default | sentence | BAAI/bge-small-en-v1.5 | |
| gpt-4o-mini | rag | dense | default | sentence | BAAI/bge-large-en-v1.5 | |
| gpt-4o-mini | rag | dense | default | sentence | thenlper/gte-large | |
| gpt-4o-mini | rag | dense | default | sentence | mxbai-embed-large-v1 | |
| gpt-4o-mini | rag | dense | default | sentence | WhereIsAI/UAE-Large-V1 | |
| gpt-4o-mini | rag | dense | default | sentence | avsolatorio/GIST-large-Embedding-v0 | |
| gpt-4o-mini | rag | dense | default | sentence | w601sxs/b1ade-embed | |
| gpt-4o-mini | rag | dense | default | sentence | Labib11/MUG-B-1.6 | |
| gpt-4o-mini | rag | dense | default | sentence | all-MiniLM-L12-v2 | |
| gpt-4o-mini | rag | dense | default | sentence | paraphrase-multilingual-mpnet-base-v2 | |
| gpt-4o-mini | rag | dense | default | sentence | BAAI/bge-base-en-v1.5 | |
| gpt-4o-mini | rag | dense | default | sentence | finance-embeddings-investopedia | |
| gpt-4o-mini | rag | dense | default | sentence | stella-en-400M-v5-FinanceRAG-v2 | |
| gpt-4o-mini | rag | dense | default | sentence | Finance-embedding-large-en-V1.5 | |
| gpt-4o-mini | rag | fusion | default | sentence | BAAI/bge-small-en-v1.5 | |
| gpt-4o-mini | rag | dense | concise | sentence | BAAI/bge-base-en-v1.5 | |
| gpt-4o-mini | react-rag-agent | dense | concise | sentence | BAAI/bge-base-en-v1.5 | |
| gpt-4o-mini | critique-rag-agent | dense | concise | sentence | BAAI/bge-base-en-v1.5 | |
| gpt-4o-mini | sub-question-rag | dense | concise | sentence | BAAI/bge-base-en-v1.5 | |

Table A6: Seeding Configurations in Experiments: Number of trials for each seeding type.

| | Experiment | Dataset | Random Seeding | Static Seeding | Transfer Learning |
|---|---|---|---|---|---|
| 1 | RAG and Agents | CRAG3 music | 100 | 46 | 0 |
| 2 | RAG and Agents | CRAG3 sports | 100 | 46 | 0 |
| 3 | RAG and Agents | DRDocs | 100 | 46 | 0 |
| 4 | RAG and Agents | FinanceBench | 100 | 46 | 0 |
| 5 | RAG and Agents | HotpotQA | 100 | 46 | 0 |
| 6 | RAG and Agents | InfiniteBench | 60 | 46 | 0 |
| 1 | Agents | FinanceBench | 100 | 3 | 0 |
| 2 | Agents | InfiniteBench | 100 | 3 | 0 |
| 1 | Seeding | HotpotQA | 46 | 0 | 0 |
| 2 | Seeding | HotpotQA | 0 | 46 | 0 |
| 3 | Seeding | HotpotQA | 0 | 0 | 46 |

## A5    Dynamic Transfer Seeding

The goal of transfer learning is to leverage experience of problem-solving on other datasets [73, 74] to speed up the search process on a new dataset. Specifically, we extract the top $k$ Pareto-optimal flows from prior runs and compute their embeddings using the BAAI/bge-large-en-v1.5 text embedding model. We then apply k-nearest neighbors (k-NN) clustering to group similar flows. From these clusters, we select a total of $N$ diverse configurations, ensuring that multiple flows from the same cluster are not chosen together. This strategy maximizes diversity in the seeded flow pool while leveraging past optimizations.

Specifically, we chose the HotpotQA dataset as the target and included the results from studies conducted on the other datasets. As a baseline, we use the standard way of starting a search from scratch: Optuna performs 10-trials of random seeding and then starts the optimization.

Fig. A3 shows a visualization of the clustering of top performing flows from these other datasets, and highlights the flows which were selected to kick-start the optimization process on HotpotQA. The results show that this form of transfer learning outperforms the static-seeding and random-seeding baselines. This result illustrates the ability to incrementally add new datasets to syftr and quickly find Pareto-optimal solutions by leveraging the results of past studies.

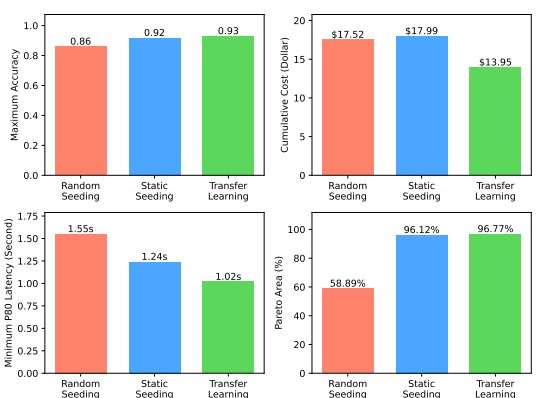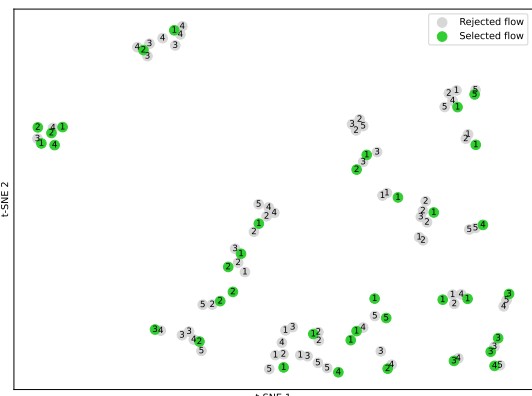

Figure A3: **Seeding Study**: (Left) Both static seeding and transfer learning outperform random seeding on HotpotQA dataset. Transfer learning from other datasets further improves latency and cost compared to static seeding. (Right) t-SNE visualization of transfer learning flows that were selected for inclusion in the HotpotQA optimization. The numbers in the dots correspond to the Pareto-frontiers (1: actual frontier, 2: next frontier after the first is removed, etc.). *K*-means clustering is applied followed by t-SNE dimensionality reduction.

## A6 Datasets

| Dataset | Sample | Train | Test | Holdout |
|---|---|---|---|---|
| HotpotQA | 20 | 7, 305 | 500 | 7, 836 |
| FinanceBench | 11 | 53 | 49 | 48 |
| CRAG3 music | 1 | 22 | 34 | 8 |
| CRAG3 sports | 7 | 39 | 46 | 11 |
| InfinityBench | 5 | 116 | 115 | 115 |
| DRDocs | 5 | 10 | 80 | 10 |

Table A7: Datasets and its partition sizes.

Table A7 gives an overview of the various datasets and dataset partitions. All dataset have `train`, `test`, and `holdout` partitions. During optimization, flows are evaluated against the question-answer (QA) pairs in the `test` partition. When the `Dynamic Few-Shot Retriever` is enabled, the `train` partition is used for finding similar examples to the query for dynamic prompt construction. Flow evaluation during optimization always uses the `test` partition. In the future, the `train` partition may also be used for LLM fine-tuning and other dynamic dataset adaptation tasks, avoiding target leakage in the `test` set. Datasets may also have a `sample` partition for development and testing purposes, which may be a separate partition or a sample drawn from the `test` set. We report accuracy numbers for flows evaluated on the `test` set, and set aside the `holdout` partition for future use.

Each dataset comes with a unique grounding data set, such as PDF, HTML, or text files which are to be used for retrieval during execution of a RAG flow. When suitable, we generate multiple partitions of the grounding data, ensuring that the required grounding data for each question is present in the appropriate partition, alongside a significant amount of 'distractor' data. This reduces the computational costs of building many large search indexes while still having a challenging retrieval problem.

**HotpotQA**: HotpotQA [52] is a large-scale question-answering dataset designed to promote research in multi-hop reasoning. It contains approximately 113, 000 QA pairs, where answering each question requires synthesizing information from multiple documents. The dataset emphasizes diverse reasoning skills, including multi-hop reasoning, comparison, and causal inference, making it a valuable benchmark for RAG flows. Each QA pair comes with one or more Wikipedia page fragments, which are used as grounding data.

We use the `train-hard` subset of HotpotQA, which has 15, 661 of the toughest questions and is split into separate `sample`, `train`, `test`, and `holdout` partitions with 20, 7305, 500, and 7836 QA pairs, respectively.

**FinanceBench**: FinanceBench [53] is a difficult RAG-QA dataset in the financial domain. The public test set includes 150 questions of varying difficulty, from single-hop qualitative lookup questions to multi-hop questions requiring complex numeric computations. It also includes 368 PDF files containing SEC filing documents from 43 companies over a seven year timespan. Answering questions using this dataset typically requires retrieving specific facts and metrics from the appropriate document by company, filing type, and time period. This is an important dataset, as it combines real-world use cases of computer-assisted financial analysis and challenges of precise information retrieval from semi-structured PDF documents, with the challenges of complex information retrieval and reasoning systems. These aspects are ubiquitous across enterprises today.

We split the dataset into roughly equal-sized `train`, `test`, and `holdout` partitions (53, 49, and 48 QA pairs, respectively), with each partition having roughly equal number of companies represented. The PDFs are also split by these partitions based on the company, so that each partition only has

PDFs from companies in the question set. This allows us to reduce the amount of grounding data in each partition, lowering the cost of optimization, while each partition still contains a significant amount of "distractor" data. The `sample` partition is drawn from the `test` partition and contains 11 QA pairs about PepsiCo. The PDF files were converted into markdown format using Aryn DocParse [56].

**CRAG**: The CRAG (Comprehensive RAG) benchmark dataset from Meta [54] was introduced for KDD Cup 2024. The AIcrowd [75] challenge contains three tasks - Task 1: retrieval summarization, Task 2: knowledge graph and web retrieval, and Task 3: end-to-end RAG. We use the Task 3 dataset only, as this is the closest task to the RAG task `syftr` is built to optimize. CRAG Task 3 (CRAG3) contains 4, 400 QA pairs on a variety of topics. The official Task 3 is to perform RAG over 50 web pages fetched from an Internet search engine for each question. We attempted a different task, which is to perform RAG over all of the web pages included in the dataset. To reduce the size of the data required for embedding and evaluation, we split the dataset into five datasets according to the five question topics - finance, movies, music, sports, and open-domain. We further partitioned each dataset into `sample`, `train`, `test`, and `holdout` partitions containing 5%, 42.5%, 42.5%, and 10% of the QA pairs, respectively.

The web page results for the QA pairs in each dataset and partition were used as the grounding data for RAG. Text from the provided HTML files was converted to Markdown format using the "html2text" [57] library.

The questions in CRAG typically contain challenging trivia about specific celebrities, events, or media, often requiring multi-hop lookup and linguistic or numerical reasoning.

Note that our task setting differs significantly from that of the official CRAG3 benchmark. We don't enforce a maximum generation time, don't restrict ourselves to Llama-based models only, and perform RAG over the entire corpus of grounding data rather than the 50 web results specific to each QA pair. Due to this, our accuracy and latency results cannot be directly compared to the contest submissions.

**InfiniteBench**: InfiniteBench [55] is a long-context reasoning benchmark dataset, containing a number of tasks including summarization, free-form QA, needle-in-a-haystack retrieval, and identification of bugs in large code repositories. We used the `En.QA` task only, which is free-form question answering based on 63 synthetically altered public domain books where character names are changed from the original. Each book consists of an average of 192, 600 tokens, for a total of 12.1M tokens of grounding data. The official task of `En.QA` is to answer questions given the entire book as context, but we use it for executing RAG over multiple books.

InfiniteBench is partitioned into `sample`, `train`, `test`, and `holdout` partitions, containing 5, 116, 115, and 115 QA pairs from 1, 22, 22, and 23 books, respectively.

**DRDocs**: The DRDocs dataset contains QA pairs about the DataRobot product suite, including GUI, API, and SDK usage, and it contains a snapshot of the entire DataRobot documentation codebase. The dataset contains 100 QA pairs, split into `train`, `test`, and `holdout` and `sample` partitions of 10, 80, 10 and 5 questions each.

## A7    Detailed Evaluation Protocol

syftr uses LLM-as-a-Judge [58] to evaluate answers generated by flows and compute the accuracy of a flow during search. The LLM-as-a-Judge compares the generated answer to groundtruth or reference answers provided in a dataset. Accuracy is a crucial metric for syftr, so it is important to understand the behavior of this evaluator since LLM-as-a-Judge can be quite sensitive to the judge prompt [58] and QA formats, often preferring outputs from their own family of models [59], simply ignoring the factual evidence and score based on the sentiment expressed in the generated answer ("vibes"), or only focusing on the final answer ignoring logical fallacies in intermediate reasoning [60]. Any such biases and variances in the LLM-as-a-Judge can be amplified when used to feedback an optimizer like MO-TPE [48].

Initially, we started by using the LlamaIndex CorrectnessEvaluator [76] with its default prompt. Here a LLM is asked to grade an answer on a scale from 1 to 5, where 4 or above is considered a passing score, and anything below 4 is failing. But due to the issues mentioned above, we decided to deeply investigate the behavior of this and other judge configurations against human judgments.

**Data Generation**: We generated 49-50 responses from each of the three datasets - CRAG open-domain, FinanceBench, and HotpotQA, using three different flows: 1. Dynamic Few-Shot prompted LLM (No-RAG flow), 2. a basic RAG flow, and 3. a ReAct RAG Agent. This generated a total of 447 individual flow responses.

We provided the query, ground truth answer, and flow response to human labelers (the authors of this work served as labelers), and asked them to evaluate the response from 1-5, where a score greater than 4 is considered a "passing" score.

The labelers were asked to prioritize accuracy relative to the provided ground truth answer from the dataset, rather than the "true" answer, in the event the provided answer differed. We do not want our judges to add their own knowledge and biases to the judgment when possible - the judges are not provided with the RAG context information and may not be aware of dataset-specific reasons for the ground-truth answer to be "wrong". The human labels were reviewed for consistent application of evaluation criteria, and several labels were modified.

**Evaluator Configurations**: We then generated evaluations for the same 447 flow responses using 10 different evaluator configurations. We used the LlamaIndex CorrectnessEvaluator, with both the default prompt and a modified prompt (see Appendix A7.1 for the templates), and tried three different LLMs with each - gpt-4o-mini, gpt-o3-mini, and anthropic-sonnet-35. This results in 6 configurations.

We also introduced a Random LLM mode and a Consensus mode. With two prompt templates each this results in 4 configurations. In Random LLM mode, each response evaluation is performed with a random selection of one of the three LLMs listed above. In Consensus mode, all three LLMs are queried and the response is labeled as "passing" if a majority of the LLMs give it a passing grade (4 or above).

Table A8 shows various statistics of the 10 different judge configurations. The experiments in Section 6 use the Default-Prompt Random LLM estimator to gain exposure to a diversity of judge LLMs without incurring the extra cost of the Consensus estimator.

### A7.1    Prompt Variations for LLM-as-a-Judge Configurations

### A7.2    Default Template

**Evaluation Guidelines for a Question Answering Chatbot**

You are an expert evaluation system for a question answering chatbot. Your task involves the following:

Table A8: Evaluator Performance Metrics relative to human judgment. The Random LLM evaluator with the default prompt template was chosen for its diversity of judges and low cost. Mean Acc Difference is the average difference in average flow accuracy for each dataset using the configured evaluator versus a human judge.

| Template | Evaluator Name | LLM | Pearson Correlation | Cohen's Kappa | Mean Acc Difference | Mean Acc Diff Std |
|---|---|---|---|---|---|---|
| Default | | gpt-4o-mini | **0.90** | 0.44 | -0.03 | 0.05 |
| | Correctness | gemini-pro | 0.76 | 0.20 | 0.09 | 0.08 |
| | | sonnet-35 | 0.86 | 0.29 | -0.05 | 0.06 |
| | Random LLM | any | 0.84 | 0.29 | **0.00** | **0.04** |
| | Consensus | all | **0.90** | 0.44 | **0.00** | 0.05 |
| Modified | | gpt-4o-mini | 0.87 | 0.49 | -0.01 | 0.06 |
| | Correctness | gemini-pro | 0.74 | 0.20 | 0.13 | 0.07 |
| | | sonnet-35 | 0.86 | 0.41 | -0.08 | 0.07 |
| | Random LLM | any | 0.83 | 0.39 | 0.01 | 0.06 |
| | Consensus | all | 0.88 | **0.48** | -0.01 | 0.05 |

**Information Provided**

- A **user query**.
- A **generated answer**.
- Optionally, a **reference answer** for comparison.

**Evaluation Task**

Your job is to judge the **relevance** and **correctness** of the generated answer. Based on your evaluation:

- Output a single score representing a holistic evaluation.
- Your score should be on a line by itself.
- Provide your reasoning for the score on a separate line.

**Scoring Guidelines**

- The score should be between **1 and 5**, where:
  - **1**: The generated answer is not relevant to the user query.
  - **2–3**: The generated answer is relevant but contains mistakes.
  - **4–5**: The generated answer is relevant and fully correct.
- If the generated answer is irrelevant, give a score of **1**.
- If the generated answer is relevant but contains mistakes, give a score of **2** or **3**.
- If the generated answer is relevant and fully correct, give a score of **4** or **5**.

**Example Response**

```
4.0
The generated answer has the exact same metrics as the reference answer,
but it is not as concise.
```

## A7.3   Modified Template

**Evaluation Guidelines for a Question Answering Chatbot**

You are an expert evaluation system for a question answering chatbot. Your task involves the following:

**Information Provided**

- A **user query**.
- A **generated answer**.
- Optionally, a **reference answer** for comparison.

**Evaluation Task**

Your job is to judge the **relevance** and **correctness** of the generated answer. Based on your evaluation:
- Output a single score representing a holistic evaluation.
- Your score should be on a line by itself.
- Provide your reasoning for the score on a separate line.

**Scoring Guidelines**

- The score should be between **1 and 5**, where:
  - **1**:
    * The generated answer mentions that the provided context does not contain all necessary information, or some important data is missing.
    * The generated answer is not relevant to the user query.
  - **2–3**: The generated answer is relevant but contains mistakes.
  - **4–5**: The generated answer is relevant and fully correct.

**Example Response**

```
4.0
The generated answer has the exact same metrics as the reference answer,
but it is not as concise.
```

## A8 Multi-Dataset Study Details

The following figures provide additional details on the Pareto-frontiers for the multi-dataset study. Fig.A4 provides a compact visualization of the Pareto-frontiers found for all datasets, while Figures A5 and A6 show expanded views of each Pareto-frontier with additional flow information.

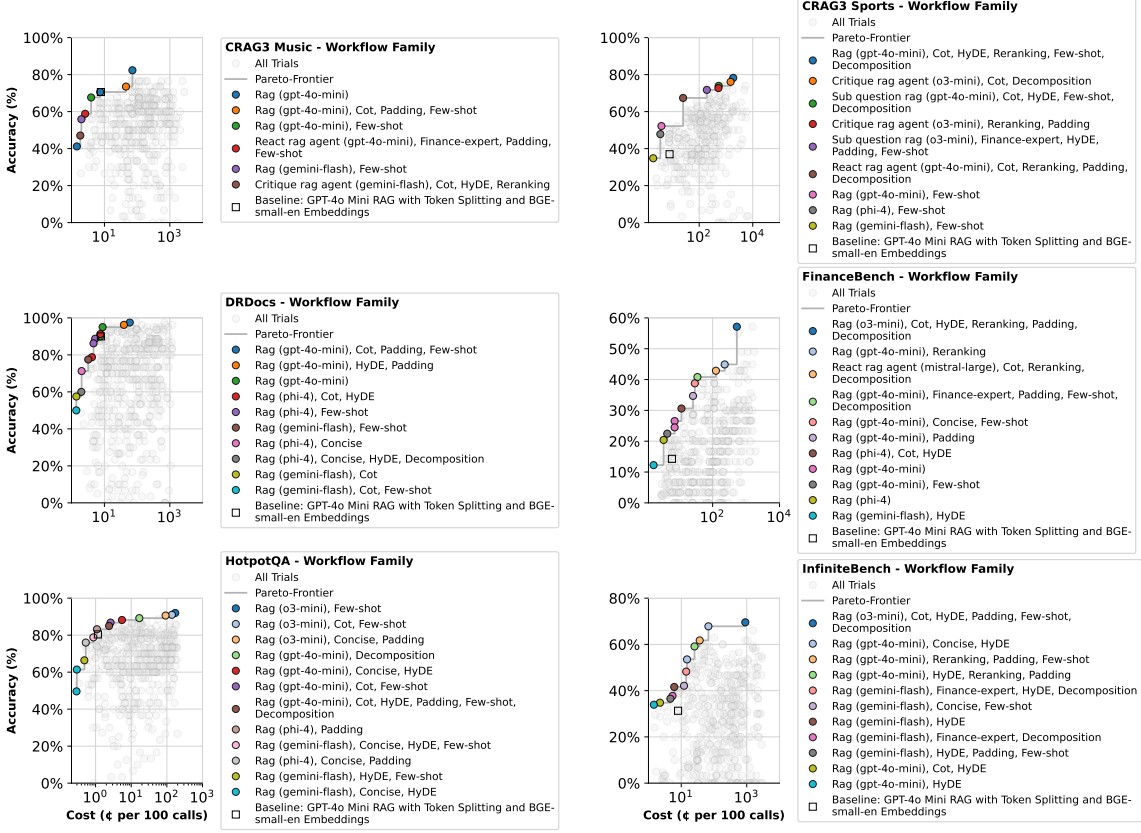

Figure A4: Multi-Dataset Study Pareto-Frontiers for all datasets.

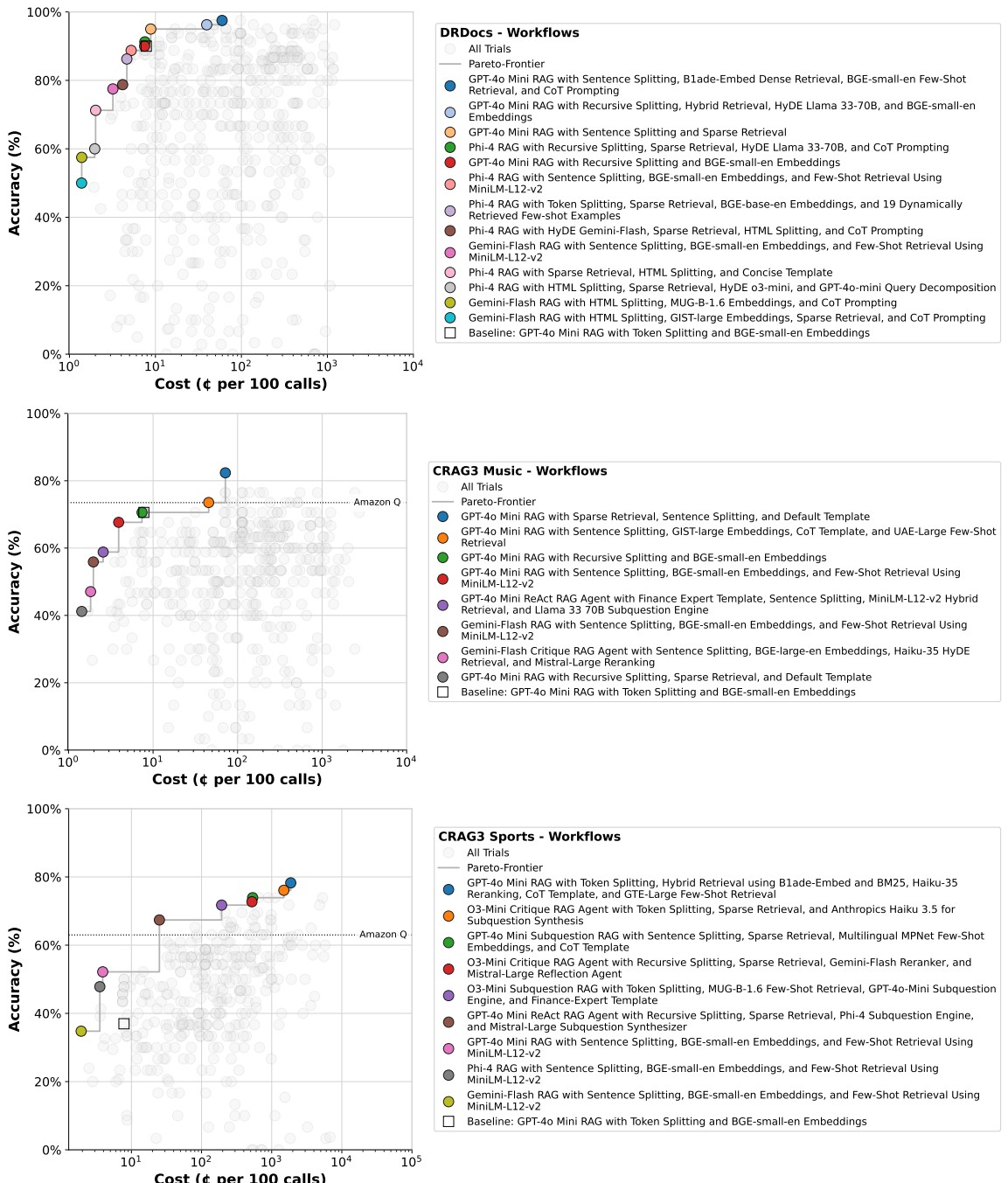

Figure A5: On DRDocs, Phi-4 open-weights model shows strong performance at various points along the Pareto-frontier, representing a notable and affordable alternative to closed weights models. On CRAG3 music, the baseline flow with GPT-4o-mini as the response synthesizer has already a good performance but there is still considerable potential to find higher accuracy or lower cost flows. On CRAG3 sports, the baseline is clearly dominated by the Pareto-flows with the potential to choose flows with roughly twice the accuracy.

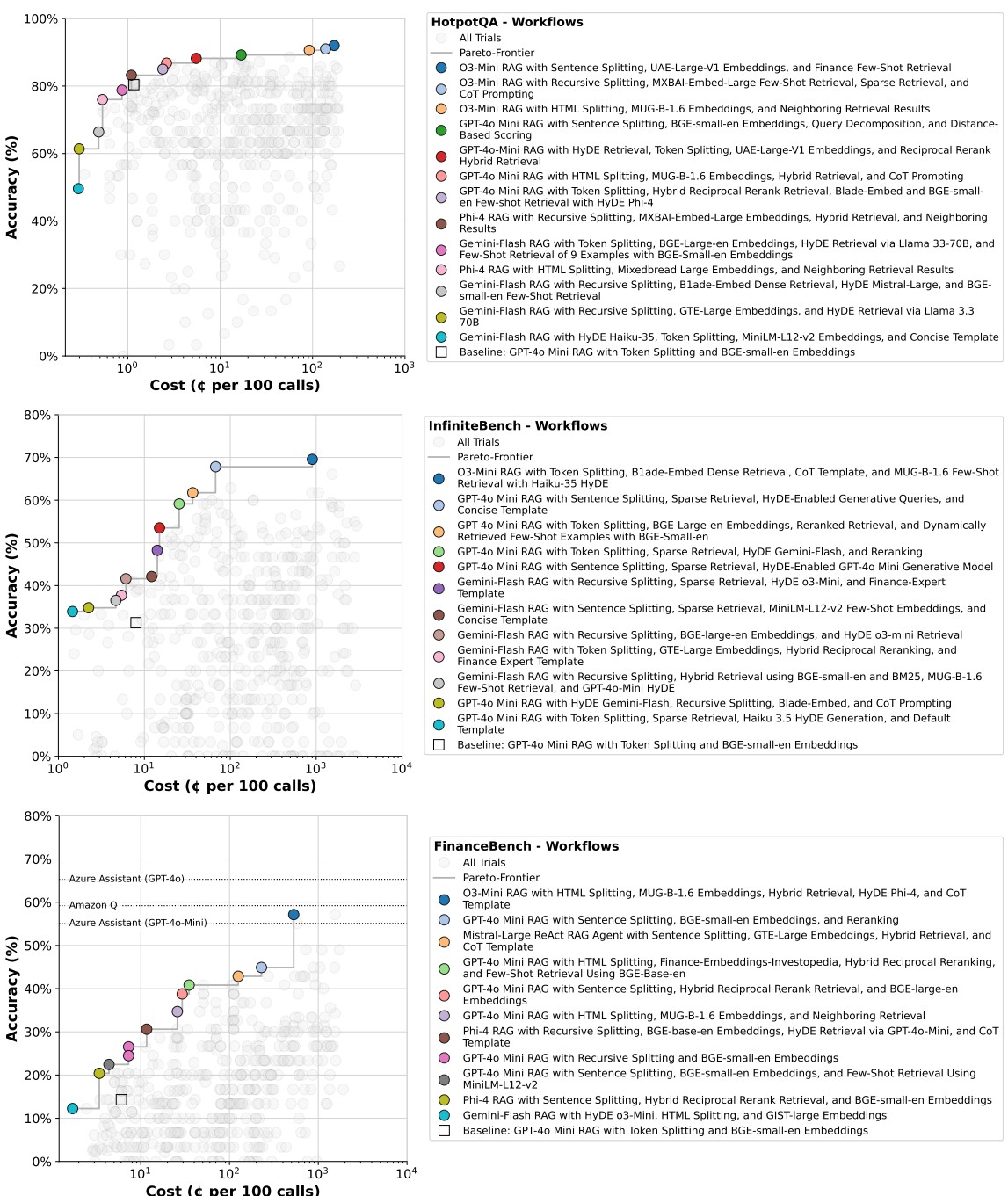

Figure A6: On HotpotQA, we see plenty of flows with a good accuracy but hugely different costs. On InfiniteBench and FinanceBench we see a wide variety of accuracies and costs spanning three orders of magnitude.

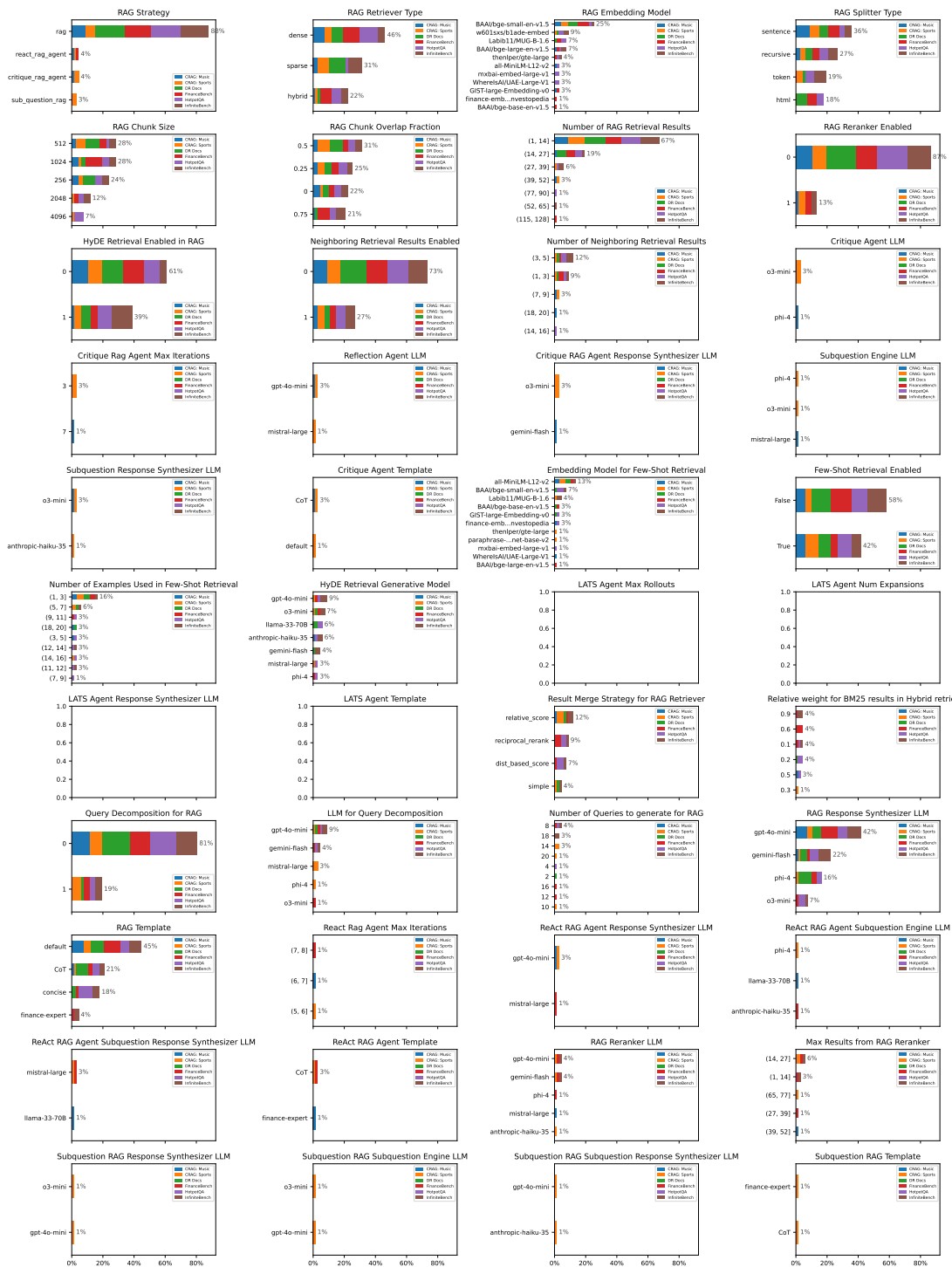

Figure A7: **Parameter Appearance**: shows the percentage of times a particular component is part of a Pareto-frontier flow across all datasets. Some insights: `Non-agentic RAG` flow is Pareto-optimal in 88% of Pareto-flows. Neighboring retrieval results is enabled in 73% of Pareto-flows. Query decompostion appears in 81% of Pareto-flows. We caution, that while such component-wise insights are useful, *how* these components are wired together as part of a larger flow matters as there are higher-order interaction effects amongst components.

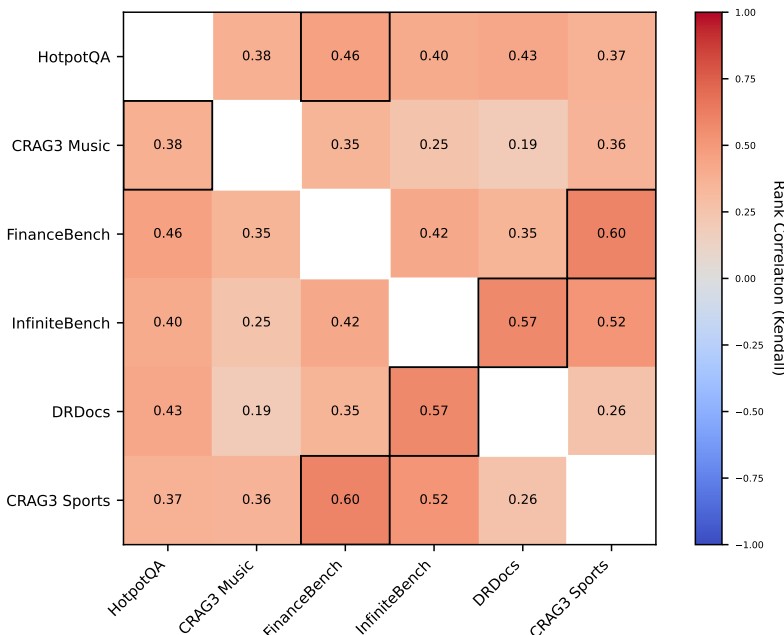

Figure A8: Rank correlation (Kendall-$\tau$) of flow accuracy across different datasets. The low correlation suggests the lack of "silver-bullet" flows that consistently perform well across diverse datasets. We hypothesize that performance of flows is highly dependent on the specific dataset characteristics, and a flow that excels on one dataset may not necessarily perform well on another. The block box indicates the highest correlation for each row.

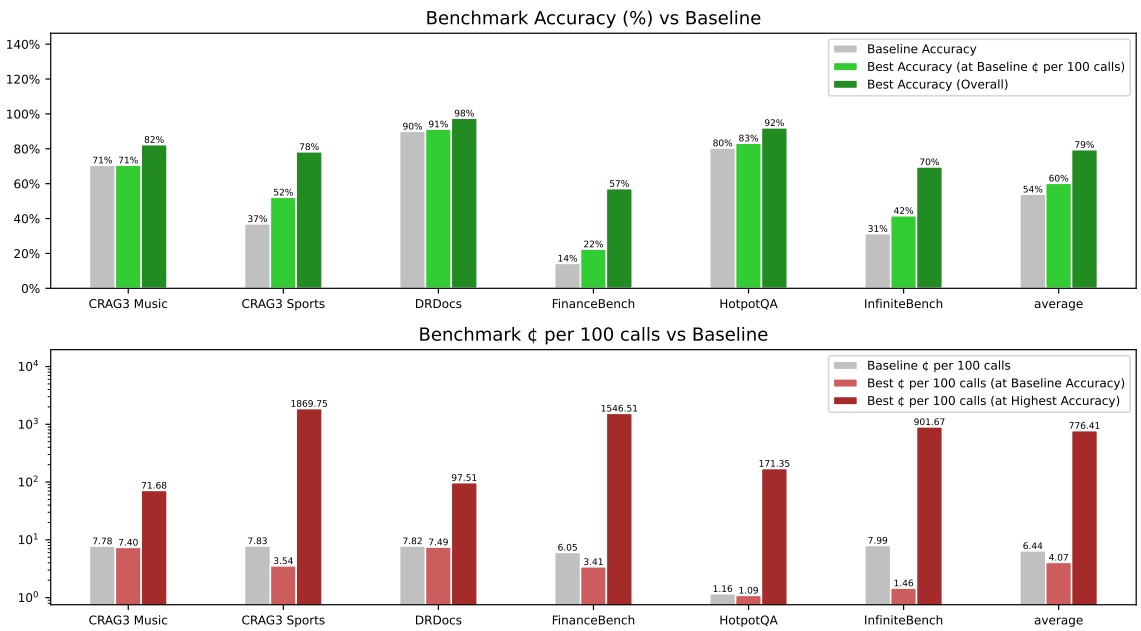

Figure A9: Baseline Comparison: across datasets `syftr` is able to identify flows that increase accuracy by 6% while retaining identical costs, or conversely decrease costs by 37% while retaining identical accuracy. If cost is no consideration, an average accuracy increase of 25% is achieved.

