# OpenReview forum: "syftr: Pareto-Optimal Generative AI"
_automl.cc/AutoML/2025/Methods_Track — AutoML 2025 Methods Track_

### Official Review · Reviewer_33TK · 2025-04-09

**Comments To Authors:**

The proposed work introduces FlowGen, a framework that employs Bayesian optimization to enhance retrieval-augmented generation (RAG) pipelines. The work aims to address the complexities of selecting among various modules and configurations to balance accuracy and cost by facilitating the creation of high-performance generative AI pipelines.
From the reviewer's perspective, the use and definition of Pareto optimality in the context is appreciated. However, some small aspects can be better discussed: (1) it would be beneficial to go deeper into the computational cost and hardware requirements associated with the proposed solution. (2) Additionally, evaluating the generalizability of the performance across various benchmarks further strengthens the robustness of the work.

**Review Confidence:**

3

**Review Rating:**

8

---

### Official Review · Reviewer_AMAM · 2025-05-03

**Comments To Authors:**

The paper presents FlowGen to craft effective RAG flows by employing multi-objective Bayesian Optimization to identify Pareto-optimal configurations, balancing accuracy and cost. It has the potential to streamline the development of high-performing generative AI pipelines tailored to dynamic and proprietary data needs.

Strength:
1) The proposed FlowGen demonstrates good engineering efforts to integrate a combination of existing techniques, such as multi-objective Bayesian optimization (i.e., the multi-objective TPE with HV contribution as the indicator for scalarization), early stop tricks, customized initialization, etc.
2) Comprehensive evaluations across many benchmarks against both academic and commercial baselines (i.e., Amazon Q)
3) Good presentation with informative figures and ablation studies.

Weakness:
1) Generalization of the searched workflows should be evaluated. Currently, every problem requires a complete execution of FlowGen, which could be practically infeasible in a real-world deployment. Do the authors identify any workflow that works reasonably well across many different problems?

2) Minor: full name should be given before its abbreviation is used, e.g., ETL.

**Review Confidence:**

4

**Review Rating:**

6

---

### Official Review · Reviewer_335f · 2025-05-11

**Comments To Authors:**

This paper introduces a framework for pipeline configuration (selection of components and hyperparameter tuning) for RAG. The framework is tested on several benchmarks.

I expect that the problem addressed becomes very relevant in the future as RAG (and agentic AI) approaches become increasingly important, and the number of available methods increases and becomes more complex. Also, as far as I know, this is a novel approach, which means that I believe that this paper has the potential to have a strong impact not only in the AutoML community but also, more generally, in AI.

METHOD

The presentation of the method is generally very clear, with most of the design decisions well motivated.

The most important issue concerns the part about seeding the optimizer: the motivation (163-167) is very good but the following choices (168-169) don't follow from it.

However, figure 2 could be improved. Namely, lines 36 and 37 are not clearly represented in the figure.

Additionally, the selection of the methods for each component of the pipeline could be justified better (page 5), although I understand this is out of the scope of this work. Maybe this could be stressed.


EXPERIMENTAL SETUP

The experiments seem to be well-designed. The method to reduce the dependence on LLM-based evaluation (191-193) is quite interesting.

I could argue that the number of problems in the empirical validation is small but this is not a limitation of this paper but of the community. In any case, it would be important that the authors acknowledge this limitation in the paper, otherwise the community tends to forget and overgeneralize the results obtained on these benchmarks.

A minor issue: is the DRDocs benchmark public?

RESULTS

The discussion of results is generally very well structured and the observations are supported by the results. In several cases, the authors speculate about the reasons for the results, which is important. But I missed the same for the key observations 1 and 3.

Additionally, was the behavior described in the key observation 2 to be expected?

One fundamental issue I would have liked to see in the discussion is the (potential) impact on the results of using models that have analyzed the data from the benchmarks. For instance, could this explain the improvements in the comparison between small vs large models (217-)?

A minor issue: the different experiments should be related to the questions in page 3.

PRESENTATION

The paper is generally well-written and is very clear. However, the analysis of results should not depend on materials in the appendices (229-231).

The main issue is the literature review, which is not well-structured: different topics are discussed without an adequate overview at the beginning and contextualization of each topic regarding the whole set and/or the previous one.

Also, the discussion of the Pareto-Pruner is not well integrated into the rest of the presentation of the method.

Additionally, a few minor issues:

-the flow between paragraphs 2 and 3 of the introduction could be improved

-(56-58) aren't 2 and the first part of 3 included in 1?

-(77) what do you mean by "holistic"?

-(109-116) formatting of the list reduces clarity

-(117-118) redundant

**Review Confidence:**

4

**Review Rating:**

10

---

### Meta-Review · Area_Chair_N3zZ · 2025-05-11

**Recommendation:** Accept
**Confidence:** 4

**Metareview:**

Given the positive reviews, I am happy to recommend this paper for acceptance. While all reviewers are on the positive side, there are some important variations on to which degree the paper is acceptable. I recommend to the authors to address the minor aspects risen by the two slightly more skeptical reviews like descriptions of hardware and naming to the camery-ready version of the paper.